# Risk Factors for Occurrence and Relapse of Soft Tissue Sarcoma

**DOI:** 10.3390/cancers14051273

**Published:** 2022-03-01

**Authors:** Pia Weskamp, Dominic Ufton, Marius Drysch, Johannes Maximilian Wagner, Mehran Dadras, Marcus Lehnhardt, Björn Behr, Christoph Wallner

**Affiliations:** Department of Plastic Surgery, BG-University Hospital Bergmannsheil, Ruhr University Bochum, 44789 Bochum, Germany; pia.weskamp@bergmannsheil.de (P.W.); dominic.ufton@rub.de (D.U.); marius.drysch@bergmannsheil.de (M.D.); johannes.wagner@bergmannsheil.de (J.M.W.); mehran.dadras@bergmannsheil.de (M.D.); marcus.lehnhardt@rub.de (M.L.); bjorn.behr@rub.de (B.B.)

**Keywords:** sarcoma, soft tissue sarcoma, relapse, risk factors, prognostic factors, sarcomagenesis, recurrence-free survival

## Abstract

**Simple Summary:**

The diagnosis and follow-up prognosis of the various number of soft tissue sarcomas (STS) subtypes are still challenging due to low incidence and variable presentation. Therefore, the demand for reliable risk factors developing STS as well as prognostic indicators for recurrence free survival remains high. The objective of this systematic review was to conduct a meta-analysis of mainly retrospective studies to assess the risk factors for development and prognostic indicators for recurrence free survival in STS for the first time. Prognostic factors determining relapse such as radiation, chemotherapy, and margins of resections as well as risk factors including smoking, genetic predisposition, toxins, and chronic inflammation were identified.

**Abstract:**

The diagnosis and prognostic outcome of STS pose a therapeutic challenge in an interdisciplinary setting. The treatment protocols are still discussed controversially. This systematic meta-analysis aimed to determine prognostic factors leading to the development and recurrence of STS. Eligible studies that investigated potential risk factors such as smoking, genetic dispositions, toxins, chronic inflammation as well as prognostic relapse factors including radiation, chemotherapy and margins of resection were identified. Data from 24 studies published between 1993 and 2019 that comprised 6452 patients were pooled. A statistically significant effect developing STS was found in overall studies stating a causality between risk factors and the development of STS (*p* < 0.01). Although subgroup analysis did not meet statistical significances, it revealed a greater magnitude with smoking (*p* = 0.23), genetic predisposition (*p* = 0.13) chronic inflammation, (*p* = 0.20), and toxins (*p* = 0.14). Secondly, pooled analyses demonstrated a higher risk of relapse for margin of resection (*p* = 0.78), chemotherapy (*p* = 0.20) and radiation (*p* = 0.16); after 3 years of follow-up. Therefore, we were able to identify risk and relapse prognostic factors for STS, helping to diagnose and treat this low incidental cancer properly.

## 1. Introduction

Soft tissue sarcomas (STS) are rare malignant tumors originating from mesenchymal or neuroectodermal stem cells, accounting for 1% of all adult malignancies. According to the WHO, there are over 100 different histological subtypes of soft-tissue tumors, the majority represented by STS, each characterized with unique clinical, therapeutic and prognostic features [1]. STS occur at the trunk, retroperitoneum and in the majority in the extremities, approximately 60% [2].

Epidemiological risk factors, including radiation and virus infections such as Epstein–Barr, are discussed to be associated with non-sporadic STS. The presence of certain genetic anomalies in which one allele of a tumor-suppressor gene is inactive due to a germline mutation are typical for the development of STS subtypes such as neurofibromatosis type I and peripheral nerve sheath tumors, just to name some [3,4]. Another hereditary genetic cancer predisposition syndrome increasing risk for sarcoma genesis is the Li-Fraumeni-syndrome, based on a germline mutation of the tumor-suppressor p53 [5,6]. Next to epidemiologic risk factors, STS develop mainly sporadically. Until now, no study has been able to define the sporadic risk factors.

The diagnosis of STS still presents a challenge to clinicians, as patients rarely show B-symptoms or paraneoplastic syndromes. Usually, STS initially become apparent due to extensive growth with consecutive pain, compression of nearby structures and following infiltration of surrounded or distant tissues. At the time of initial diagnosis, 10% of STS have already metastasized hematogenous into the lung. Postmetastasis survival remains under 15 months for patients with STS, highlighting the importance of timely diagnosis [7]. By determining STS risk factors, the differential diagnosis of this low incidence cancer can be made faster and easier when identifying typical clinical symptoms.

Surgical resection presents the mainstay of treatment. Contrary to initial assumptions, amputation does not increase overall survival in adult patients with STS of the extremities when compared to limb-salvage surgery [8]. It is the complete surgical resection with negative surgical margins that provides the potential for curative treatment and is at the same time recognized to be the most important prognostic factor of local recurrence-free survival (LRFS) [9].

Moreover, operation time has been shown to be an independent predictor of wound complications, while tumor-size is not only associated with surgical complications, but also increases rates of local recurrence [10]. Another predictor of poorer local control is described in the proximity between STS and major blood vessels [11].

The low incidence combined with the difficulty of variable presentation, characterization and behavior of the subtypes highlight the importance of centralized care and specialized treatment. Interdisciplinary oncologic decisions are made with the teamwork of radiologists, pathologists, radiotherapists, medical oncologists as well as surgical oncologists [12]. The diagnosis of STS can be made with the combination of on MRI with pre-treatment biopsy, mainly punch biopsy, and subsequent histologic analysis [13].

The aim of this study was to investigate risk factors associated with STS as well as prognostic indicators of LRFS to further its general understanding of sarcoma genesis and etiology. Therefore, we conducted a systematic review and meta-analysis of mainly retrospective cohort studies and a few randomized controlled trails (RCT) that determined potential risk factors.

## 2. Materials and Methods

### 2.1. Search Strategy

In preparation for the search, a preliminary review of the literature was performed to determine the characteristics and quantity of published literature including retrospective cohort studies and RCT that assessed risk factors for STS and predictors for relapse free time as well as survival time. Broad search strategy for different data bases such as PubMed/ MEDLINE, EMBASE and Cochrane were used. The Cochrane and PRISMA guidelines for the conduct of systematic reviews were followed for this study. We have registered the study at OSF with the following link: doi:10.17605/OSF.IO/DAEWK.

### 2.2. Identification of Eligible Studies

A systematic search was conducted by two independent investigators to identify controlled trials and retrospective cohort studies in which different risk factors for soft tissue sarcoma and predictors for relapse free time (RFT) and survival time was researched and discussed until January 2021. The search was not limited by language, endpoint of study, blinding, geographical origin, study design and sponsorship. The presence of the following criteria was assumed: 1. the patients studied were of any age with the occurrence of an STS; 2. the participants underwent treatment; 3. as a main outcome, postoperative survival rate was defined; 4. analyses of risk factors for STS, predictors for RFT and survival time were performed.

To determine eligible studies according to the inclusion criteria, two independent researchers validated the titles and abstracts of studies identified by the search criteria. Articles that did not feature all the aforementioned criteria during the initial research remained potentially eligible and were retrieved in full text and reviewed to ensure eligibility. Here, disagreements were discussed with a third reviewer until consensus was found. Moreover, a references list of eligible studies was analyzed to identify missed potentially eligible studies. Finally included studies and their characteristics are presented in Table 1. Figure 1 summarizes the PRISMA search strategy.

### 2.3. Data Extraction and Bias Assessment

For the extraction of study data, a standardized pre-designed data extraction form was used. Data were collected from each eligible randomized controlled trails and retrospective cohort study independently by two authors. The following data were extracted: authors, publication date, study design, multi- or single-center, sample size, purpose of study, STS site, neoadjuvant and adjuvant therapy (radio- and chemotherapy). Two review authors independently assessed risks of bias using the Cochrane ‘Risk of bias’ tool. We assessed several study characteristics for risks of bias, including random sequence generation, allocation concealment, blinding, incomplete outcome data, selective outcome reporting and other potential sources of bias. Based on these criteria, we rated the studies as having a low, high or unclear risk of bias for each category. We discussed any disagreements about risks of bias, and in cases of remaining discrepancies, a third author was consulted for consensus.

### 2.4. Statistical Analysis

The study data was analyzed with Review Manager 5 (RevMan5). For dichotomous outcomes, we calculated the odds ratio (OR) with a 95% confidence interval (CI). We assessed heterogeneity using the chi2 test and the I2 statistic. Here, an I2 value of less than 25% indicates low heterogeneity, greater than 50% moderate and greater than 75% high heterogeneity.

For the chi2 test, we considered a *p*-value of 0.05 to be statistically significant. If the I2 statistic and chi2 test suggested heterogeneity, we visually inspected the forest plot for outliers. We used a sensitivity analysis (e.g., excluding outliers) to explore potential explanations for heterogeneity.

### 2.5. Risk of Bias in Included Studies

Using the Cochrane risk of bias tool, methodological quality of trials is visualized in Figure 2 and Figure 3. Most of the studies were assessed with low risk of bias, some with unclear concerns but no study with high risk of bias was included. Moreover, a funnel plot was used to evaluate publication bias between the studies which is presented in Figure 4.

## 3. Results

### 3.1. Study Population

Final eligible studies comprised 6452 patients. The sex ratio male to female was 42 to 48 percent while the median age at diagnosis was 51 years (range, 18–99 years). The most common histological STS subtypes were liposarcomas, leiomyosarcomas, malignant fibrous histiocytofibromas, and undifferentiated sarcomas with histological grades ranging from one to three. Tumors were in the upper and lower limbs, chest wall and head. All included trials mentioned different risk factors for STS such as smoking, genetic predispositions, toxins, and chronic inflammation. These have been analyzed statistically and visualized in Figure 5.

### 3.2. Smoking, Genetic Predisposition, Toxins and Chronic Inflammation Trigger the Development of Soft Tissue Sarcoma

An overall effect of developing STS with a greater and statistically significant magnitude was found in the test of overall studies. Therefore, smoking, chronic inflammation, toxins and genetic predispositions are comprehensively associated with the development of STS (*p* < 0.01). Analysis of each subgroup effect revealed a greater magnitude but did not meet statistical significance. Developing STS with cigarette smoking history (odds ratio (OR): 1.1796 (*p* = 0.23), genetic predisposition (odds ratio (OR): 1.2419; (*p* = 0.13) chronic inflammation, (odds ratio (OR): 1.1407, (*p* = 0.20), as well as the effect of toxins (odds ratio (OR): 1.2348; (*p* = 0.14) showed no singularly significant effect. Significant heterogeneity among the included studies was not found (Chi^2^ = 16.47, I^2^ = 0%). The pooled data analysis is presented below in Figure 5.

### 3.3. Chemotherapy and Radiation Decrease the Risk of Recurrence

Secondly, the effect of chemotherapy, radiation, and margins of resection on the local recurrence was analyzed. The meta-analysis was conducted on fixed effect model (*p* = 0.06). Additionally, pooled analyses demonstrated a lower risk of relapse for chemotherapy treatment (*p* = 0.20) as well as radiation therapy (*p* = 0.16), after three years. Subgroup analysis of the margin of resection showed a greater effect (*p* = 0.78). An overall effect was found (*p* = 0.14) although it did not meet statistically significance. The data presented in below Figure 6.

## 4. Discussion

This meta-analysis represents the first study addressing the correlation between predisposing risk factors for the occurrence of STS as well as factors associated with LRFS. We were able to demonstrate that smoking, genetic predisposition, toxins, and chronic inflammation show an increased risk of developing STS, while chemotherapy (CTx), radiation therapy (RTx) and negative surgical margins are associated with a decreased risk of local recurrence.

### 4.1. Sarcomagenesis

#### 4.1.1. Smoking

Smoking is associated with pathogenesis of multiple malignancies, while simultaneously negatively impacting disease outcome [16]. Furthermore, studies have shown an overall increased cancer mortality risk regardless of gender [17]. For STS, smoking has been shown to reduce distant metastasis-free and progression-free survival, although no adverse effects on overall survival could be determined [18]. A different study reported an association between smoking and increased risk of death in patients under the age of 50, leading the authors to conclude that smoking prematurely ages patients prognostically [19]. However, no study has been conducted regarding the direct association between smoking and sarcomagenesis.

Cigarette smoke contains more than 4000 separate components, including toxins, oxidants, and direct carcinogens [20]. The heterogenous components lead to pro-inflammatory and carcinogenic effects, while simultaneously acting immunosuppressive. In addition to depletion of macrophages, B-cells and cytotoxic T-cells, smoking reduces the amount of circulation natural killer (NK) cells [21,22]. As NK-cells are responsible for continuous tumor immune surveillance, this reduction can partially explain increased occurrence of malignant tumors in smokers. Cigarette smoke could be shown to increase tumor burden in a murine lung metastasis model following melanoma B16-MO5 challenge [23]. For a different murine model, it was reported that lymphocytes protect against the development of carcinogen-induced sarcomas [24].

As previously mentioned, the impaired NK-cell-dependent tumor surveillance and other immunosuppressive effects are only partially responsible for the smoke-induced increase and acceleration in development of tumors. Inflammatory mediators found in cigarette smoke induce growth factors through the phosphoinositide 3 kinase (PI3K)-AKT-pathway, which is involved in apoptosis and proliferation, promoting tumor cell proliferation by influencing multiple downstream signaling pathways including mTOR, NF-κB and MDM2, a negative regulator of p53 [25]. The PI3K-pathway is negatively regulated by PTEN. In a mouse model, an upregulation of the PI3K-AKT-pathway by inactivation of PTEN in smooth muscle cell lineage resulted in significantly increased occurrence of leiomyosarcoma [25,26]. Moreover, cigarette smoke also leads to chronic inflammation due to increased serum levels of IL-6, IL-8 and CRP [27].

Therefore, we can conclude that prolonged cigarette smoke exposure induces the development of STS in humans by decreasing immunity and increasing burden of carcinogenic mediators. Taken together, the content from basic research regarding oncogenesis and our results suggest smoking to be a facilitating agent in sarcomagenesis if not primarily responsible in some cases.

#### 4.1.2. Chronic Inflammation

Chronic inflammation itself is a well-established component of tumor progression [28,29], the origin of up to a quarter of all cancer cases can be related to chronic inflammation and infections [30]. Several types of chronic inflammation, differing in cause, mechanism and intensity have been shown to stimulate cancer development and progression: Prolonged exposure to cigarette smoke, leading to chronic respiratory inflammation increases the risk of lung cancer [31]. Furthermore, obesity has been identified as a risk factor for developing hepatocellular carcinoma via increased production of IL-6 and TNF [32].

Even with cancers that have no epidemiological association to specific inflammation types, oncogenesis is maintained by inflammatory mediators that contribute to genetic instability and the proliferation of malignant cells [33]. There is evidence of a synergism between pre-malignant or fully malignant cells and inflammatory cytokines produced by tumor infiltrating immune cells. Here, it was shown that survival was promoted, angiogenesis was induced and advancing metastasis of cancer cells was advanced [34,35]. In a retrospective study including 103 patients with synovial sarcoma, pre-treatment inflammatory indexes were independent prognostic factors of overall survival and progression-free survival [36].

In 2019, Dadras et al. demonstrated the oncologic impact of wound complications after curative resection of primary soft tissue sarcomas of the chest wall [37]. Patients with wound complications had an estimated 5-year local recurrence-free survival (LRFS) of 30% versus 72.6%, and a 5-year disease-specific survival (DSS) of 58.3% versus 82.1%. Therefore, wound complications could be identified as an independent predictor for worse LRFS and DSS.

Li et al. (2016) contributed significant prognostic value of systemic inflammatory markers including CRP and neutrophils to lymphocytes ratio (NLR) in pre-operative blood in bone and soft tissue sarcoma patients. They demonstrated a significantly higher risk of recurrence and overall decreased survival rates in sarcomas when higher level of pre-operative CRP and NLR are present [38].

Driving factors linking inflammation to oncogenesis and especially sarcomagenesis include signaling pathways activated by primary inflammatory cytokines such as IL-1, IL-6, IL-8, TNF and prostaglandins [39]. Pre-treatment serum levels of IL-1, IL-6, IL-8, IL-10 and TNF-alpha were significantly elevated in patients with STS when compared to healthy subjects [40]. Therefore, determining serum IL-6 levels may be used as a diagnostic tool to differentiate between benign soft tissue tumors and STS [41].

On a molecular level, transcription-factors and signaling pathways associated with inflammation, such as HIF1alpha, STAT3 and NF-kB, show correlation with sarcomagenesis and progression. Therefore, PI3K-AKT-, Ras-Raf-MAPK-, sonic hedgehog- and the Notch-pathway are considered to be the driving forces behind inflammation-induced sarcomagenesis [42]. That is why understanding the role of inflammation in sarcomagenesis identifies not only diagnostic measures but also presents possibilities for novel therapeutic approaches. Recent studies targeting signaling-pathways related to chronic inflammation in STS have shown promising results. The inhibition of NF-kB pathways with dehydroxymethylepoxyquinomicin resulted in reducing proliferation while inducing apoptosis in osteosarcoma [43]. Moreover, for human liposarcoma cells, a chemo sensitizing effect of the semisynthetic flavonoid 7-mono-O-(β-hydroxyethyl)-rutoside cells by limiting NF-kB induction by doxorubicin was reported [44].

In conclusion, chronic inflammation plays an overall important role in STS development, therapy, and prognosis.

#### 4.1.3. Toxins

The role of toxins, especially dioxin, in sarcomagenesis remain a controversial topic, with few studies reporting differing results [45,46,47,48]. Dioxins are categorized as the so-called persistent organic pollutants. As by-products of combustion, incineration, and several industrial processes, they are usually not produced purposely. High level of dioxins emissions can be observed from municipals, hospitals, and hazardous chlorine-containing waste incineration. They are also formed during the production of some halogenated organic chemicals.

Zambon et al. (2007) stated a 3.3 times higher risk of developing a sarcoma for subjects with long and high exposure levels of dioxin caused by waste incinerators and industrial sources of airborne emissions in a part of the Province of Venice [46], whereas Benedetti et al. (2020) showed no increased risk of STS resulting from industrial incinerator emissions in their case–control study of subjects living close to the Mantuas district of Italy [48].

#### 4.1.4. Genetic Predisposition

Multiple inherited genetic disorders predispose to sarcoma development. These include Li-Fraumeni syndrome (LFS), neurofibromatosis type I, hereditary leiomyomatosis, renal cell cancer and retinoblastoma [3,49,50]. In LFS, germline mutations in the p53 tumor suppressor gene are proven, whereas somatic mutations in the p53 gene are observed in 30% to 60% of STSs [51,52]. As a transcription factor, the p53 protein usually inhibits cell growth and at the same time stimulates cell apoptosis [53,54]. If DNA is damaged, p53 up-regulates p21, which binds to cyclin complexes and temporarily stops cell division. In cases of severe DNA damage, p53 activation is prolonged and then stimulates cellular death [55,56]. Most of the studied p53 mutations result in loss of function that block this important pathway to apoptosis, although there exist certain mutations that may induce gain of function, which is still not well investigated [57]. As studies report, p53 mutation may be an early event in STS development with complex nondiscrete gene alternations [58,59]. Telomeric dysfunction and impaired joining of non-homologous chromosomal ends may lead to p53 inactivation and trigger cellular defense disability against tumorigenicity.

A key function in sarcoma development may be germline or somatic genetic alternations. It is noteworthy that increasing understanding of the oncogenic mechanisms underlying human sarcomagenesis is being generated due to rapidly evolving high throughput of genomic and proteomic technologies [59]. Fusion genes, which are sarcoma subtype-specific, may be especially useful as molecular targets for diagnosis and treatment. However, knowledge about the complex cellular and molecular mechanisms that regulate sarcomagenesis is still vestigial, but it can be a starting point for further studies about the molecular basis of sarcoma development, proliferation, and dissemination.

### 4.2. Relapse-Free Survival

Numerous analyses have been conducted to assess the prognostic factors that affect LRFS in patients with STS. Among these factors, next to histological grade, tumor size, depth, and histological subtype, future episodes of local recurrence, margins of resection and postoperative treatment with radiation or chemotherapy are the most significant [13,60,61]. Another interesting prognostic factor was examined in the Dadras et al. study (2020) where wound complications have been identified to be an independent predictor of local recurrence and disease specific death of primary STS of the chest wall [37].

Although patients with STS of the extremities still die due to distant metastasis, local tumor control presents the only adjustable treatment factor for non-metastasized STS, making improvement of this aspect pivotal to STS treatment. Zaho et al. and Eilbar et al. were able to report reduced 5-year OS for high-grade STS patients with local recurrence compared to those who did not develop recurrent disease [11,12]. Our results show a significant and independent association of margin of resection, radiation, and chemotherapy with LRFS greater than 36 months.

#### 4.2.1. Margin of Excision

Limb-sparing surgical resection with clear margins is considered the therapy of choice. An escalation of therapy could result in considerable impairment of extremity function, particularly in cases of large tumor size or localization adjacent to critical anatomic structures [62]. A complete excision must be the primary goal as multiple studies with large patient populations have proven that positive microscopic resection margins significantly decrease LRFS [60,61,62,63,64]. Due to the close connection of margin of resection and local recurrence, relapse can be viewed as a marker of quality of previous local treatment. It is noteworthy that solely the quality of the surgical margin influences prognosis, whereas the width of the surgical margin is irrelevant as long as fully negative resection margins have been achieved [64]. Henceforth, the aim of therapy should be negative resection margins, while simultaneously maintaining functionality of the affected limb whenever feasible.

In a study conducted by Harati et al., resection margin was not only significantly associated with LRFS and DSS, but also MFS [64]. A possible explanation is the capability of locally recurrent STS to metastasize and consecutively negatively impact DSS. However, the role of local recurrence in distant metastasis remains a controversial topic with multiple studies reporting contradicting results [61,65,66]. Determining whether local recurrence signifies progression towards metastatic disease, or merely acts as a sign of aggressive tumor-biology, still needs to be done. Nevertheless, quality of surgical margins independently predicts local control and OS. In a retrospective analysis of 181 STS-related deaths, 17% of patients after R1-resection died of locally recurrent disease without prior development of distant metastasis, highlighting the independent detrimental impact of local recurrence on OS [67].

#### 4.2.2. Radiotherapy

When limb-salvage surgery is feasible, neoadjuvant or adjuvant radiation therapy plays a large role in the perioperative management. The goal of radiation therapy is improvement of local control rates and functional outcome. Our findings regarding the improvement of local control via radiation therapy are congruent to the results of an earlier meta-analysis by Strander et al., who condone the use of adjuvant radiation therapy for patients with STS of the trunk and extremities to achieve local control after limb-salvage surgery [68]. It remains unclear whether pre- or postoperative admission of radiation therapy is favorable regarding LRFS. An RCT comparing preoperative to postoperative radiation therapy showed no significant difference on local recurrence, although OS was slightly higher in the preoperative group [69]. A 2010 meta-analysis reported improved local outcome for preoperative radiation; however, these results did not reach statistical significance when the random effects method was used [70]. Preoperative radiation therapy has been shown to greatly improve LRFS for certain radiosensitive STS-subtypes, especially myxofibrosarcoma, myxoid liposarcoma, vascular sarcoma [71,72].

Another form of radiation therapy is brachytherapy, in which a postoperative irradiation is performed, after intraoperative placement of catheters within the surgical bed. Multiple studies have shown adjuvant brachytherapy to reduce local recurrence rates in patients with high-grade STS [60,73].

Radiation therapy should also be discussed regarding to preselection. Guidelines recommend the indication for radiation therapy in the case of a more complicated course or larger findings. Ultimately, this leads to a negative bias in the selection of patients for radiotherapy.

#### 4.2.3. Chemotherapy

Compared to surgery and RTx, CTx plays a limited role in the treatment of STS. As different STS-subtypes differ in chemosensitivity, choosing an adequate therapy regiment poses a challenge and usually remains a case-by-case decision in the interdisciplinary management of STS.

In this study, we were able to show a greater association with relapse-free survival over three years after CTx (OR 1.28). These findings coincide with the meta-analysis of Pervaiz et al. from 2008, including 1700 patients and 296 cases of local recurrence. They reported a significant decrease in local recurrence with an OR of 0.73 in favor of chemotherapy. Statistical significance was shown, when combining all 17 studies in statistical analysis, whereas subgroup analysis for adjuvant doxorubicin-based therapy and adjuvant Doxorubicin-based therapy combined with Ifosfamide yielded no statistically significant results [74]. An absolute risk reduction (ARR) of 3% was shown for a doxorubicin-based CTx alone, while the ARR was 4% when analyzing all eligible trials with a number-needed-to-treat of 25 [74]. These results are in line with the Sarcoma Meta-Analysis Collaboration from 1997, which reported an increased LRFS with an OR of 0.73 and an ARR of 6% (improving from 81% to 75%), pooling data from 13 trials including 1315 patients [75]. For both meta-analyses, patients receiving varying doses of doxorubicin may have impacted the results. Research data for the effects on local recurrence with therapy regiments not including doxorubicin remain scarce. Therefore, reasons such as heterogeneity in chemosensitivity of different STS-subtypes, identification of suitable high-risk populations and overall rarity of STS present difficulties for conducting high-evidence research [76]. Overall, the results of the meta-analyses coincide with our findings. Yet, in a recent RCT comparing standard CTx (anthrazyklin + ifosfamide) to histotype-tailored CTx regiments, no significant benefit regarding local control could be shown for either group at 46 months of follow-up [74].

### 4.3. Limitations of This Study

The limitations of these studies should be noted. The meta-analysis was based on data extracted from literature and could suffer from some possible bias, although no obvious bias was found in funnel plots. The patient populations across the included studies were heterogeneous with respect to specific soft tissue sarcoma pathologic subtype, treatment therapy as well as tumor location. We performed tests for heterogeneity between studies for all endpoints and did not find significant heterogeneity regarding the STS risk factors indicating the appropriateness of data pooling for these studies [77], at the same time admitting that weak heterogeneity was found among the studies pointing out the prognostic relapse factors in their single analyses but not in the overall test heterogeneity. Moreover, we conducted subgroup analysis to find the source of heterogeneity and assessed their influence on the results.

One of the weaknesses is the lack of a formal analysis according to histology, since we could not subdivide groups by histology in the meta-analysis. In fact, most studies reported STS development in mixed populations without separating the histological types. Although this is the first meta-analysis defining risk factors for STS development in general, we recommend further research should focus on different histologic subtypes of STS and their risks factors specifically.

It must be stated that no single subgroup analysis meets significance criteria (*p* < 0.05), but the overall test effect of STS risk factors was significant (*p* < 0.01). Therefore, we can deduce that there must be an association between the STS development and the sum of the identified risk factors. Moreover, it is the first study to analyze these risk factors, so we can still postulate clinical use and an improved pre-test probability in finding the diagnosis of STS.

## 5. Conclusions

In this study, we were able to show that the combination of smoking, genetic predisposition, toxins, and chronic inflammation is associated with an overall significantly increased risk for the development of STS. Including these factors in diagnosis finding, the patients’ outcome may improve due to early therapy. Furthermore, we were able to underscore the prognostic relevance concerning local STS recurrence of chemotherapy, radiation therapy and margin of resection. Therefore, prognostic patient evaluations may lead to improved prediction of survival rates and other clinical outcome parameters. At the same time, inclusion of these parameters as potential correlative variables in clinical trials and cohort studies allow further stratification of patient population within future studies. Moreover, additional basic research regarding sarcomagenesis and STS progression, as well as the treatment options increasing LRFS, present an opportunity to develop and improve prognostic scores, diagnostic measures, and treatment options.

## Figures and Tables

**Figure 1 cancers-14-01273-f001:**
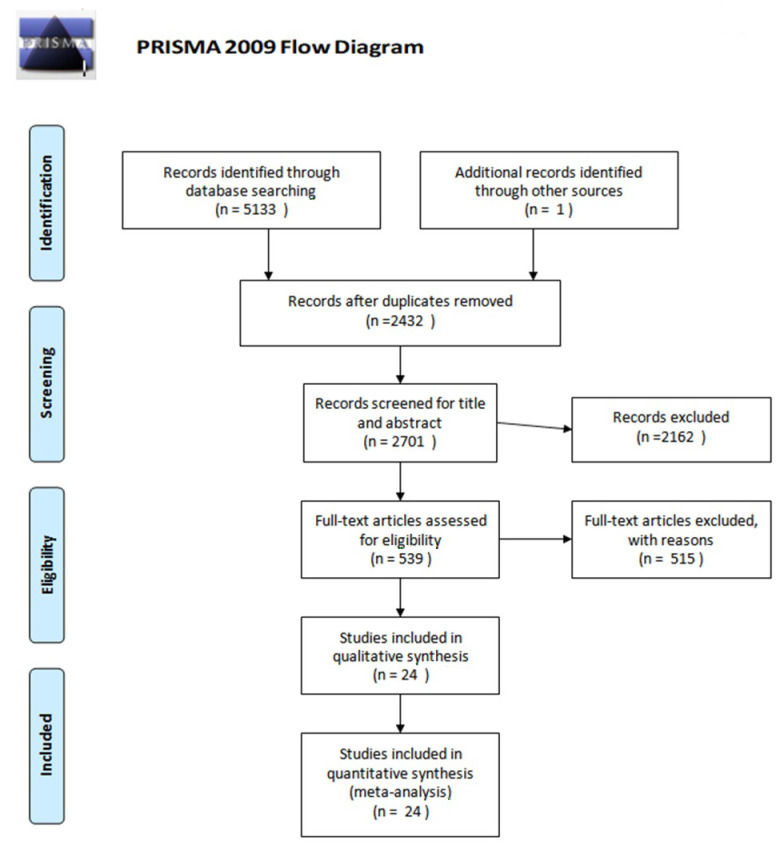
Study flow diagram by The PRISMA Group (2009) [14]. 24 studies were included in the meta-analysis.

**Figure 2 cancers-14-01273-f002:**
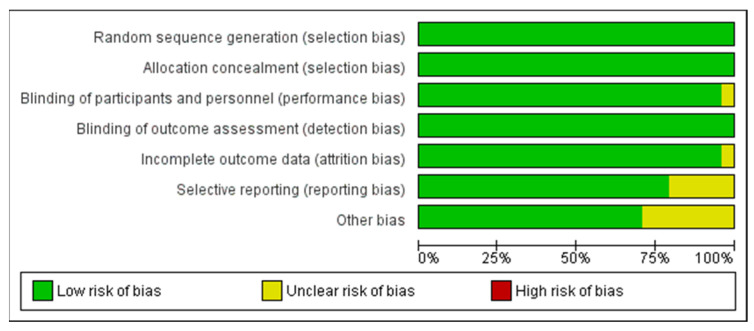
Mainly low risk of bias and no high risk was found in the included studies. We used the Cochrane ‘Risk of Bias’ Tool RoB2 [15].

**Figure 3 cancers-14-01273-f003:**
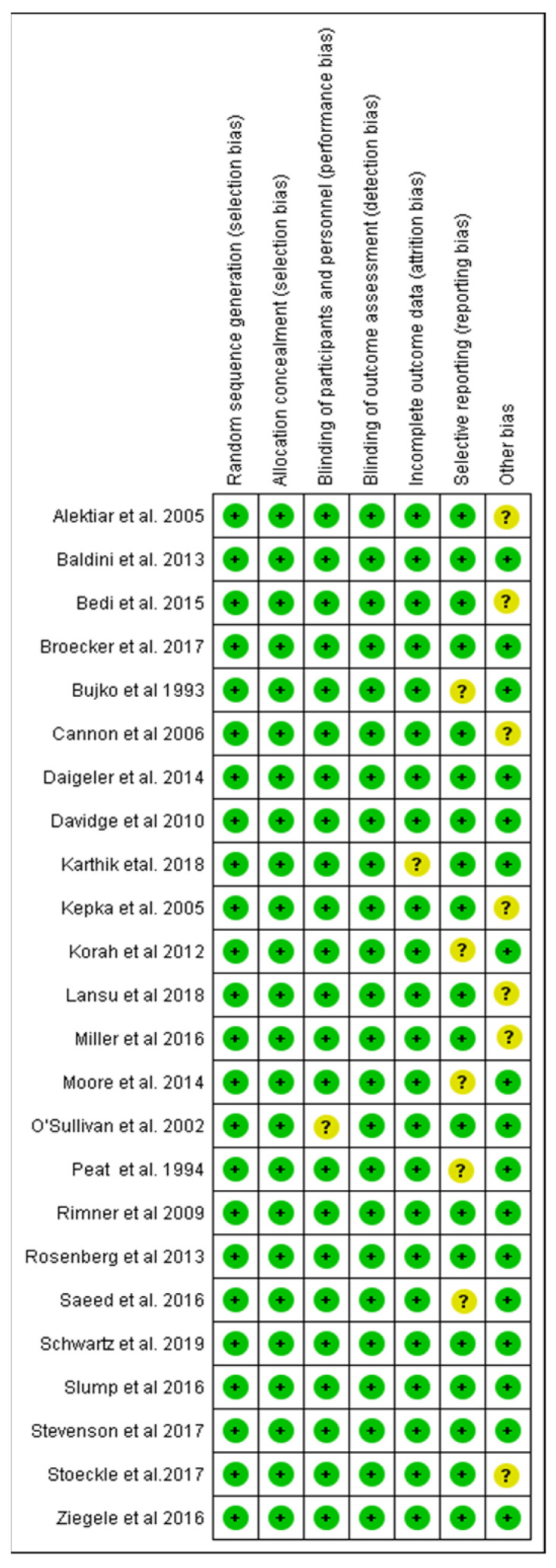
Symbols in the traffic light plot show low (green, +), unclear (yellow, ?) and a high risk of bias (red, -). This figure again was created by the risk of bias tool RoB2, provided by Cochrane [15].

**Figure 4 cancers-14-01273-f004:**
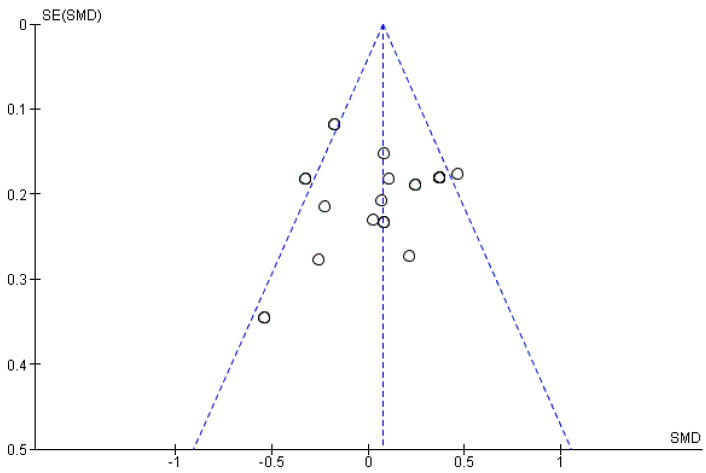
Funnel plot showing no significant publication biases among the included studies. SMD = Standardized Mean Difference; SE = Standard Error.

**Figure 5 cancers-14-01273-f005:**
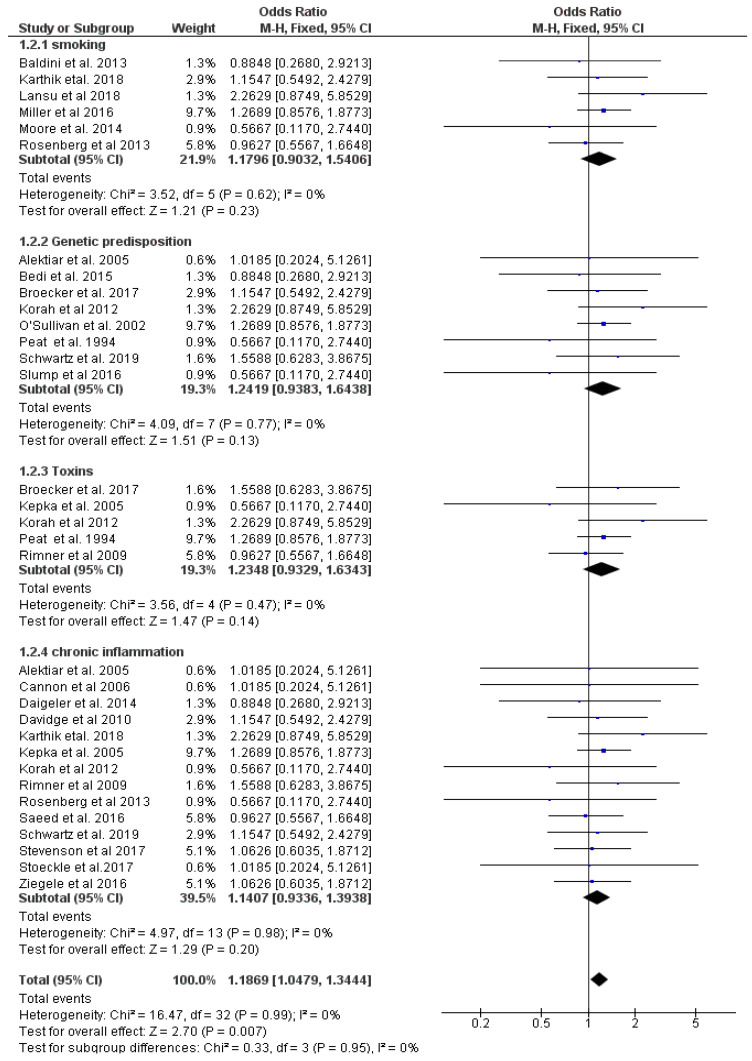
Pooled data for risk factors developing STS: smoking, genetic predisposition, toxins, and chronic inflammation. The OR is seen as aberration from the vertical line. Therefore, an OR > 1 increases the risk of STS genesis.

**Figure 6 cancers-14-01273-f006:**
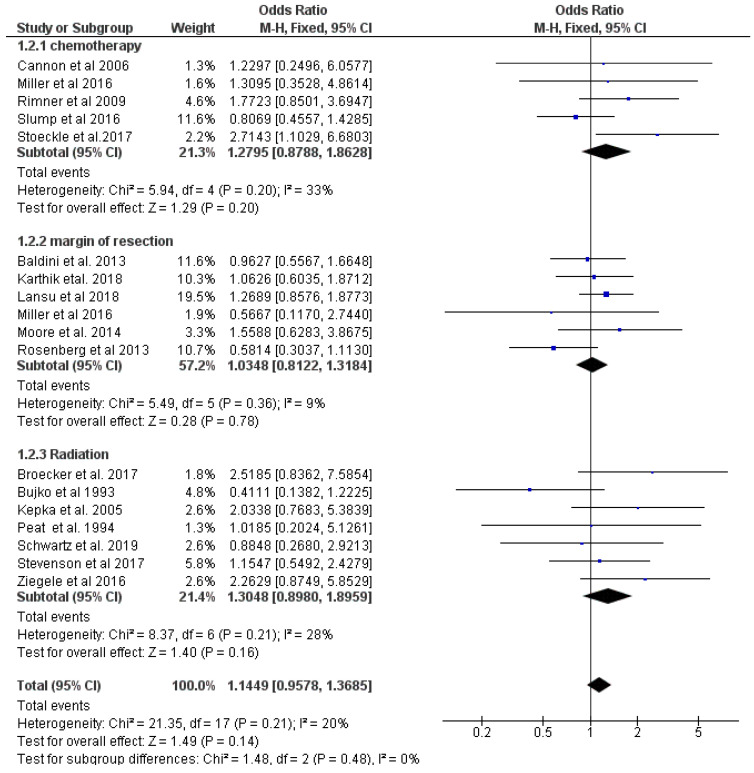
Forest plot of data for relapse free survival rate, greater than 3 years. The OR is seen as aberration from the vertical line. Therefore, an OR > 1 increases the relapse free survival rate.

**Table 1 cancers-14-01273-t001:** Included studies presented in the data extraction form.

AUTHORS/YEAR	STUDY DESIGN	CENTER	NUMBER OF PATIENTS	STS SITE	RTX PREOP IN %	RTX POSTOP IN %	CTX IN %
BUJKO ET AL. 1993	retrospective	single center	202	Upper and lower extremities/Head/Trunk	100	71	0
PEAT ET AL. 1994	retrospective	single center	180	Upper and lower extremities/Trunk	40	100	0
KEPKA ET AL. 2005	retrospective	single center	112	Upper and lower extremities/Head/Trunk	100	0	21 preop
BALDINI ET AL. 2013	retrospective	multi center	103	Upper and lower extremities/Trunk	100	0	18 preop
MOORE ET AL. 2014	retrospective	single center	256	Upper and lower extremities/Head/Trunk	48	24	8 preop, 8 postop
BEDI ET AL. 2015	retrospective	single center	92	Upper and lower extremities/Trunk	100	0	38 preop
SAEED ET AL. 2016	retrospective	single center	245	Upper and lower extremities/Trunk	71	14	28
BROECKER ET AL. 2017	retrospective	single center	546	Upper and lower extremities/Trunk	35	10	23 preop, 10 postop
STOECKLE ET AL. 2017	retrospective	single center	728	Upper and lower extremities/Trunk	0	70	28 preop
KARTHIK ET AL. 2018	retrospective	single center	271	Upper and lower extremities/Trunk	16	24	17 preop
O’SULLIVAN ET AL. 2002	randomized controlled trail	multi center	182	Upper and lower extremities/Trunk	92	94	0
ALEKTIAR ET AL. 2005	retrospective	multi center	369	Upper and lower extremities	0	100	34 postop
CANNON ET AL. 2006	randomized controlled trail	single center	416	Lower extremities	65	35	41
RIMNER ET AL. 2009	retrospective	single center	255	Thigh	0	100	31
DAVIDGE ET AL. 2010	retrospective	multi center	247	Upper and lower extremities	62	7	0
KORAH ET EL. 2012	retrospective	single center	118	Upper and lower extremities	81	19	29
ROSENBERG ET AL. 2013	retrospective	single center	73	Upper and lower extremities	100	8	18
DAIGLER ET AL. 2014	retrospective	single center	135	Upper and lower extremities/Trunk	0	27	3
ZIEGELE ET AL. 2016	retrospective	single center	81	Thigh with pelvis	86	4	31
MILLER ET AL. 2016	retrospective	single center	102	Upper and lower extremities	25	75	39
SLUMP ET AL. 2016	retrospective	single center	897	Upper and lower extremities	54	6,1	5,4
STEVENSON ET AL. 2017	retrospective	single center	127	Upper and lower extremities	45,7	54,3	0
LANSU ET AL. 2018	retrospective	single center	191	Upper and lower extremities	100	0	1,5
SCHWARTZ ET AL. 2019	retrospective	multi center	571	Upper and lower extremities/Trunk	12	0	15 preop, 16 postop

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
