# Peer review of "Risk Factors for Occurrence and Relapse of Soft Tissue Sarcoma"

_cancers, 2022, doi:10.3390/cancers14051273_

Round 1
Reviewer 1 Report
Thank you for the opportunity to review this article. The article analyses potential risk factors for STS occurrence and recurrence. The topic is interesting and the publication of these data could potentially be useful to readers, but I perceived too many methodological mistakes or shortcomings. Moreover, the conclusions are not at all consistent with the results. I report a list of considerations that deserve a revision by the authors:
1) the introduction is too long and some additional reference would be welcome; for example regarding the topic of recurrences I recommend: "Sambri, A., Caldari, E., Montanari, A., Fiore, M., Cevolani, L., Ponti, F., D'Agostino, V., Bianchi, G., Miceli, M., Spinnato, P., De Paolis, M., & Donati, D. M. (2021). Vascular Proximity Increases the Risk of Local Recurrence in Soft-Tissue Sarcomas of the Thigh-A Retrospective MRI Study. Cancers, 13(24), 6325. https://doi.org/10.3390/cancers13246325”
2) You said: "The objective of this systematic review was to conduct a metaanalysis of randomized controlled trials (RCTs) to assess the risk factors for development and prognostic indicators for recurrence free survival in STS for the first time.” Not only RCTs were analysed in this review, but they are a small minority of the pool of articles analysed. Please correct these inaccuracies everywhere in the text. For example, in the Methods section you said: In preparation for the search, a preliminary review of the literature was performed to determine the characteristics and quantity of published literature including prospective and retrospective RCTs that assessed risk factors for STS and predictors for relapse free time as well as survival time.”. Again, what are retrospective RCTs? RCT is an acronym for randomised controlled trial and identifies a precise type of study design. In fact in section 2.2 you say correctly: "A systematic search was conducted by two independent investigators to identify controlled trials and retrospective cohort studies in which different risk factors for soft tissue sarcoma and predictors for relapse free time and survival time was researched and discussed until January 2021.”
3) Regarding table 1:
- the purpose of the studies is stated in a very confusing manner, the use of abbreviations is excessive and they are not made explicit in the text, making them effectively not understandable;
- I would replace "both extremities" with "upper and lower extremities",
- the columns "radiotherapy" and "chemotherapy" are impossible to understand, the use of brackets seems variable, and often not specified, these columns should be simplified or their number increased to facilitate the understanding of the data reported.
4) although no reference is reported, figure 1 suggests that the PRISMA 2009 guidelines were used, which are now really outdated. Please enclose the completed PRISMA 2020 protocol checklist in the resubmission, clarifying the details of all sections. Of course figure 1 will also have to be modified accordingly.
5) lines 115-120 are unnecessary as these data should be provided in figure 1.
6) For the assessment of study quality and risk of bias you have used the Cochrane 'Risk of bias' tool. Please provide a reference. In case the tool used was validated for the assessment of clinical trials only, it will be necessary to repeat the quality assessment using a checklist that allows to assess also retrospective cohort studies or other study designs (if present among the articles analysed in this review). Furthermore, the results of the reported analysis seem to be definitely too optimistic considering the amount of retrospective and low evidence studies included.
7) Figure 5 and Figure 6 are difficult to understand as it is not shown which direction of the graph represents increased risk and which represents decreased risk.
8) In the results you reported the odds ratios as percentages. In addition, the p-value is sometimes approximated to the first number after the decimal point, sometimes to the second, sometimes to the third. Such warnings cause diffidence in the statistical validity of the results and the methodological soundness of the analysis. Please provide the raw data of the individual articles or the intermediate data of the analysis as supplementary material.
9) In discussion you said: "We were able to demonstrate that smoking, genetic predisposition, toxins, and chronic inflammation are associated with an overall significantly increased risk of developing STS, while chemotherapy (CTx), radiation therapy (RTx) and negative surgical margins decrease the risk of local recurrence”. This is not true since no independent risk factors with statistical significance have been identified. It is possible to say that some factors are associated with an increased risk of occurrence or recurrence of STS, although without any claim to significance. This is a misleading manner of presenting the results on by the authors.
10) In discussion you said: “It must be stated that no single subgroup analysis meets significance criteria (p< 0.05). But since the overall test effect of STS risk factors was significant (p = 0.007) we can still assume prognostic effect of these STS development indicators in total. Moreover, it is the first study to analyse these risk factors, so that we can still postulate clinical use and an improved pre- test probability in finding the diagnosis of STS.”. To attribute so much weight to a concept such as 'if all possible risk factors occur together in the same patient, they significantly increase the risk of occurrence' is wrong in my opinion. Please remove this consideration or soften its relevance in the text.
11) In Conclusions you said: “In this study, we were able to identify smoking, genetic predisposition, toxins, and chronic inflammation as factors independently associated with a significantly increased risk for the development of STS.” This is absolutely false, for two reasons:
- these risk factors are only significant if they are all present together;
- it is not possible with the available data to establish their role as independent factors as they are not deduced from multivariate analyses.
Please revise the conclusions to make them consistent with the results.
Thank you.
Author Response
See the letter attached

Reviewer 2 Report
In the review “Risk factors for occurrence and relapse of soft tissue sarcoma”, the authors performed a systematic meta-analysis to determine the risk factors leading to soft tissue sarcoma (STS) development and recurrence. Among the investigated studies, they identified smoking, genetic predisposition, chronic inflammation and toxins as prognostic factors for STS development. Margin resection, chemotherapy and radiation demonstrated to be correlated with risk of relapse.
The work by Weskamp et al. is exhaustive and well written. My concerns are the followings:
- The manuscript is presented as a review. However, its structure is organized as an original paper, including materials and methods, results and discussion sections.
- The prognostic factors identified as leading to development and recurrence of STS are risk factors commonly correlated to tumorigenesis. I am wondering whether some risk factors more specifically linked to STS can be identified.
- In the meta-analysis, the range of patients analyzed is from 18 to 99 years old. Why did the authors exclude patients under 18 years old from the study? Some STS specifically affect children or adolescence.
- Figure 5 is smaller than the others. Its dimensions should be increased.
- Figure legends should be more detailed.
Author Response
See letter attached

Round 2
Reviewer 1 Report
Thanks for the changes. The authors have addressed most of my concerns.
Only a few considerations:
1) lines 420-421: the sentence seems to me unreadable, please reword it keeping the meaning;
2) lines 425-427: please further clarify that the sentence refers to all risk factors together;
3) lines 428-430: please specify that you are now talking about recurrences.
Thank you.
Author Response
See letter attached
